# Soil Fungal Pathogens in *Pinus pinaster* Mature Reforestation: Silvicultural Treatments Effects

**DOI:** 10.3390/pathogens13080637

**Published:** 2024-07-30

**Authors:** Iciar Martos, Jose Alfonso Domínguez-Núñez

**Affiliations:** Department of Forest and Environmental Engineering and Management, Universidad Politécnica de Madrid, C/Jose Antonio Novais, 10, 28040 Madrid, Spain

**Keywords:** thinning, gap, forest diversity, fungi, soil pathogen, plant disease, Spain

## Abstract

Soil fungal communities play a key role in multiple functions and ecosystem services within forest ecosystems. Today, forest ecosystems are subject to multiple environmental and anthropogenic disturbances (e.g., fire or forest management) that mainly lead to changes in vegetation as well as in plant-soil interactions. Soil pathogens play an important role in controlling plant diversity, ecosystem functions, and human and animal health. In this work we analyzed the response of soil plant pathogenic fungi to forest management in a *Pinus pinaster* reforestation. We started from an experimental design, in which forest thinning and gap cutting treatments were applied at different intensities and sizes, respectively. The fungal communities of plant pathogens in spring were described, and the effect of the silvicultural treatments was evaluated 5 years after their application, as were the possible relationships between soil plant pathogenic fungal communities and other environmental factors. Only a strong low thinning treatment (35% basal area) was able to generate homogeneous changes in soil pathogenic diversity. In the gaps, only the central position showed significant changes in the soil plant pathogenic fungi community.

## 1. Introduction

Fungal communities play a key role in driving multiple functions and ecosystem services in forest ecosystems, including organic matter decomposition (e.g., organic phosphorus and nitrogen mineralization), water regulation, and nutrient release and uptake by trees (Baldrian, 2017 [1]). At the same time, fungi can contribute to more than 80% of the microbial biomass in these ecosystems (Joergensen & Emmerling 2006 [2]) and can account for up to one-third of soil C (Clemmensen et al., 2013 [3]). Today, forest ecosystems are subject to multiple modes of environmental and anthropogenic disturbances (e.g., climate change, pests, fire, and forest management) that primarily drive changes in their vegetation component (e.g., vegetation cover, tree density, residual wood dynamics; Baldrian, 2017 [1]), as well as in plant-soil interactions. However, despite the importance of forest fungi, we know relatively little about how disturbances in the forest may influence the diversity, productivity, and functioning of soil fungal communities (Kohout et al., 2018 [4]).

In forests, canopy gaps are crucial elements for forest dynamics and diversity (Liu & Hytteborn, 1991 [5]). Root pathogens can appear in and on the fine roots of older trees, compromising the survival of new congeneric seedlings growing in new gaps close to these trees (Chavarriaga et al., 2007 [6]). Thus, soil pathogens play an important role in controlling plant diversity, ecosystem functions, and human and animal health (Bever et al., 2015 [7]; Ciccarone & Rambelli, 2000 [8]). They alter plant competition and community composition (Delgado-Baquerizo et al., 2020 [9]; Dinoor & Eshed, 1984 [10]). The risk of pathogen attack on plants varies depending on environmental conditions, which affect pathogen and plant fitness (Barrett et al., 2009 [11]). Also, environmental factors, including regional (precipitation and temperature) and local (plant and soil) factors, have important influences on soil pathogen diversity (Delavaux et al., 2021 [12]; Delgado-Baquerizo et al., 2020 [9]).

Previous work has documented contrasting results regarding soil fungal responses to forest management. In relation to fungal diversity, in general, the effect of thinning appears to be negative; a decrease in species richness and/or changes in fungal community composition may occur (Baral et al., 2015 [13], Lin et al., 2015 [14], Maghnia et al., 2017 [15], Müller et al., 2007 [16]). On the other hand, retention silviculture (leaving “parent” trees) may also be more positive for fungal diversity compared to “clear-cutting” (Luoma et al., 2004 [17], Rosenvald and Lohmus 2008 [18]); however, these positive effects may be due to environmental changes induced by site-level forest management, with localized effects (Varenius et al., 2017 [19]). Other silvicultural proposals to increase fungal diversity focus on improving stand structural complexity (Dove and Keeton 2015 [20]). The impact of removing woody debris from the forest remains unclear: some studies have reported negative effects on the productivity of important marketable fungal species, such as Boletus edulis (Salerni and Perini, 2004 [21]), while other authors have reported the opposite trend for ectomycorrhizal species (Baar, J., Ter Braak, C.J.F., 1996 [22]).

All these observations emphasize the need to determine the real effect of forest management operations on soil fungal communities in the short and medium term. Studying the effects of disturbances on soil fungal communities is important for predicting changes in soil chemistry and fertility. It is also important to understand how disturbances (such as silviculture) shape fungal communities according to their characteristics (Treseder and Lennon, 2015 [23]).

Within the trophic group of soil pathogenic fungi there is little relative literature reporting on the specific effects of silvicultural treatments. We hypothesize that forest management practices modify soil-borne plant pathogenic fungal communities in the short to medium term by modifying soil environmental conditions, dispersal conditions, and the presence of potential hosts. The present work aims to evaluate the effect, after 5 years, of two silvicultural treatments (thinning and gap cutting) on the soil-borne plant pathogenic fungal community of a mature *P. pinaster* reforestation pine forest; in addition, we will analyze their possible relationship with other environmental factors.

## 2. Methodology

### 2.1. Area and Experimental Design

The study was carried out in public forest 261 of Jócar (Guadalajara, Spain) (Figure 1). It corresponds to a Koppen’s Csb climate (temperate with a dry and mild summer) and is in the supramediterranean bioclimatic floor with a subhumid ombroclimate. The average annual precipitation is 652 mm, and the average annual temperature is 11.5 °C. According to the FAO classification (1988 [24]), the main soil groups are umbrisols and alisols. The average bulk density of the soil is 1.45 ± 0.09 g cm^−3^ and its texture is loamy loam (sand = 30.9 ± 2.7%, silt = 51.3 ± 1.8% and clay = 17.8 ± 3.6%). The pH is 5.32 ± 0.13 and the electrical conductivity is 0.34 ± 0.06 dS m^−1^, while the organic matter concentration is 1.45 ± 0.50%, nitrogen is 0.07 ± 0.01%, phosphorus is 7.84 ± 0.86 mg kg^−1^, and potassium is 75.4 ± 18.0 mg kg^−1^. The dominant vegetation in the work area is a pine forest of *Pinus pinaster* from reforestation carried out in 1965, which is described below. Under the pine forest there is also a variable woody understory of trees such as oak (*Quercus ilex* subsp. ballota), gall oak (*Quercus faginea* Lam.), and juniper (*Juniperus oxycedrus* L.), and shrubs such as the following: *Salvia rosmarinus* Spenn, *Cistus ladanifer* L., *Cistus laurifolius* L., *Daphne gnidium* L., *Rosa* sp., *Crataegus monogyna* Jacq., *Erica scoparia* L., *Erica arborea* L. and *Rubus ulmifolius* (Oliet et al., 2022 [25]).

Within the scope of the Spanish project Forest Management for Adaptation and Mitigation FORADMIT (AGL2016-77863, 2017–2020), an experimental design was established during the winter of 2017–2018, in which two major silvicultural experiments were defined: a thinning experiment and a selection system of cutting groups or gaps (Figure 1). The thinning experiment had three blocks with four levels each, where four thinning intensities were randomly distributed: (1) control (no thinning); (2) moderate low thinning treatment (20% basal area removal; B20); (3) strong low thinning treatment (35% basal area removal; B35); and (4) strong selective thinning treatment (35% basal area removal, A35). Each square plot of thinning measured 50 × 50 m. Part of the cut trees were randomly left as a source of coarse woody debris in half of each thinned plot (subplot). Tree density in the control area was 539 ± 14 trees. The ha^−1^ and basal area was 36.2 ± 0.8 m^2^ ha^−1^. These logs were randomly left on the ground to evaluate the impact of coarse woody debris on different ecological processes. The other half of each plot lacked accumulated coarse residual wood. The gap experiment had three blocks, and each block had three levels: (1) three control plots (no cutting); (2) nine “small” gaps (cutting in a 26 m diameter circular gap, Ø26, 1.5 times the dominant height (Ho = 17.5 m) of the stand: 530.9 m^2^); and (3) three “large” gaps (cutting in a 44 m diameter circular gap, Ø44, 2.5 times Ho: 1520.5 m^2^). More details of the experimental design can be found in Oliet et al. (2022 [25]).

As to soil sampling areas, in the thinning stand, transects (50 m × 4 m) were proposed along each thinning half-plot (with or without residual wood), and in the three blocks: each transect was positioned in the middle of each half-plot except for the control transect, which was placed in the middle of the plot; in the gap stand, three gaps (1/block) were randomly selected of each diameter; three circular subplots (bottom edge–center–top edge) were also considered in each selected gap; each subplot measured 4 m diameter (Ø). In this way we tried to analyze a hypothetical effect on pathogen communities by slope and maximum insolation (with and without canopy shade) within each gap (Appendix A). The direction of the two subplots with respect to the center of the gap was always set as SE-NW, which was the intermediate direction between the average direction of maximum slope (E–W) and the direction of maximum insolation without canopy shade (S–N); the gap controls were established in forest subplots.

### 2.2. Soil Sampling and Processing

Soil sampling was conducted during the spring of 2023. In the thinning plots, approximately 10 soil cores (approx. 2 cmØ × 15 cm depth each, according to soil conditions; 4 Soil Tip Kit DB-4 Oakfield Apparatus^®^, Oakfield, WI, USA) were zigzag sampled every 5 m and mixed along each thinning semiplot transect (4 × 50 m; Appendix A); in the gap plots approx. 5 similar soil cores were randomly sampled and mixed in each 4 mØ circular subplot. In summary, a total of 42 composite samples were finally obtained; soil samples were stored at 4 °C for <24 h and further processed.

Each composite soil sample was homogenized and sieved (Ø 3 mm); a 50 mL subsample of each sample was freeze-dried, reduced to a fine powder by mortar, and stored in a dark and dry environment until the following DNA extraction process.

### 2.3. DNA Extraction, NGS Library Preparation, Sequencing and Analysis of ITS2 Sequencing Data

Genomic DNA was extracted from 500 mg of soil using the NucleoSpin^®^ soil NSP kit (Macherey-Nagel, Duren, Germany). A quality control of the DNA sequences was performed by means of Qubit^®^ Fluorometer. For libraries preparation, the “16S Metagenomic Sequencing Library Preparation” manual from Illumina was followed adapted to fungi analysis. Briefly, the ITS2 region of the fungal genome was amplified using primers that contain gITS7 and ITS4 sequences (Ihrmark et al., 2012 [26]) and Illumina specific adaptors. In a second amplification, Illumina sequencing adapters and dual-index barcodes were added to the amplicons. Libraries were quantified with the NEB Next Library Quantification kit and pooled equally. DNA libraries were paired-end sequenced (2 × 300) on an Illumina NextSeq 1000/2000 system with the use of the NextSeq™1000/2000 P1 Reagents (600cycles) and following the protocol recommended by the manufacturer. DNA Library preparation and sequencing was performed at the Unidad de Genómica of the Universidad Complutense de Madrid. The FASTQ files containing the sequencing reads were analyzed using the CLC Genomics Workbench version 20.0.4 (QIAGEN Aarhus A/S www.qiagenbioinformatics.com). Sequence data were trimmed using 0.01 as a limit for quality scores with 2 as the maximum number of ambiguities. The reads after trimming were analyzed using the CLC Microbial Genomics Module version 4.0. The optional merge paired reads method was run with default settings. Sequence reads were clustered and chimeric sequences detected using an identity of 97% as the Operational Taxonomic Unit (OTU) threshold. Reference OTU data used in the present study were downloaded from the UNITE database v.9.0 for fungal rDNA. The functional identification of the taxa identified to species or genus level was performed by cross-referencing the taxa with the FUNGALTRAITS database (Põlme et al., 2020 [27]).

### 2.4. Environmental Data

For each of the 42 sampling areas (21 transects and 21 circular subplots) within the overall experimental area, we considered macroclimatic conditions to be homogeneous; adjusted average environmental data for each one of these sampling areas (insolation, slope, altitude, aspect and vegetation cover) were extracted using ArcMap 10. 8.1© (spatial analyst tools), from the Digital Terrain Model (PNOA MDT05, MDT25) and derived products (MDP05; NDSM-vegetation), and from public use data of the National Geographic Institute of Spain https://www.scne.es/ (Appendix A). The most current vegetation cover information available corresponded to the period of 2019–2020.

It must be taken into account that the insolation data correspond to the radiation value that would reach the first layer of vegetation, different to the one that would reach the ground directly.

### 2.5. Data Analysis

All statistical analyses were performed for the group of plant pathogenic fungi. The relative abundance of pathogens was defined as the proportion of OTU sequences. For the study of communities, sequences with a relative abundance higher than 0.5% were considered. The diversity function of the Past 4.03 software package (freeware data analyzer app) was used to estimate diversity by calculating richness (number of OTUs), Shannon diversity, Simpson diversity and evenness. The significance of the differences in relative abundance, Shannon and Simpson diversity, richness, and evenness of plant pathogenic fungi for the different treatments, intensities, and sampling positions were evaluated using the Statgraphics19© statistical package; the normality of the data was checked, ANOVA analysis was performed, the homogeneity of variances was checked using Levene’s test, and Tukey’s or Duncan’s mean comparison test was performed (*p* < 0.05); the Kruskal-Wallis test was applied in the case of non-parametric statistics.

Bray-Curtis’s dissimilarity was analyzed for fungal communities using the NMDS (Non-metric multidimensional scaling) ordination method. This analysis was performed using Past 4.03 software. The effects of the different sites on fungal communities were analyzed by multivariate analysis of variance with permutations (PERMANOVA; 9999 permutations).

Redundancy analysis (RDA) is a method to extract and summarize variation in a set of response variables (e.g., pathogen community structure, diversity indices, or abundance of major genera) that could be explained by a set of environmental factors. Therefore, RDA was performed to explore significant relationships between plant pathogen communities, diversity parameters, and major genera and environmental factors via Past 4.03. Also, relationships among fungal diversity, major pathogenic fungal genera, and environmental factors were examined using Pearson’s and Spearman’s correlations. The environmental data were normalized before representation and analysis. The environmental variables were checked for pairwise correlation and one variable of each pair with a Pearson correlation coefficient > 0.7 was discarded. In this case we discarded the aspect variable, as it was strongly correlated with insolation (R^2^ > 0.97).

## 3. Results

### 3.1. Abundance of Soil Pathogens

The distribution of relative abundances of soil fungal pathogens showed a higher frequency of plant and animal pathogenic fungi in almost all sites (Figure 2), and a scarce presence of mycoparasites and lichen parasites. In any case, the total number of fungal pathogens never exceeded 18% of the relative abundance of the total soil fungal community. A great irregularity of abundances was observed between blocks, although this factor did not significantly affect (*p* > 0.05) the relative abundance of plant pathogenic fungi. In no case did it exceed 12% in any of the sites. In addition, average relative abundances ranged from 1.5% (gap ends Ø44) to 6% (selective thinning A35) (Figure 3). No interactions between site factors were observed. When comparing total plant pathogenic fungi (Figure 3a), a significantly higher average relative abundance (4.7%) was observed in the thinning than in the gaps (2.4%), for *p* < 0.05. However, within each treatment, the relative abundances of plant pathogens were not affected by the intensity of thinning or the presence of coarse woody debris in the thinning plots, nor by the diameter or sampling position in the gap’s plots (Figure 3).

### 3.2. Diversity of Soil Plant Pathogens

For the diversity parameters (Figure 4), no interactions between site factors were observed. Overall, mean plant pathogen species richness (Figure 4a) was significantly higher (*p* < 0.05) in the thinning plots (19) relative to the gap’s plots and control (15); however, mean evenness (Figure 4d) was significantly higher (*p* < 0.05) in the gaps (0.17) relative to the thinning (0.07). Shannon and Simpson indices did not reflect significant differences between the two treatments of thinning and gaps. Within thinning plots, neither thinning intensity nor the presence of coarse woody debris significantly affected richness or Shannon or Simpson indices. However, B35 showed significantly greater evenness (0.17) (*p* < 0.05) than A35 (0.07). The presence of coarse woody debris (CMM) seemed to show more dispersed (but not significant) values of diversity with respect to SMM; B35 increased Shannon and Simpson indices (non-significantly) and evenness (significantly) the most. In the center of the gaps, a significant (*p* < 0.05) increase of all diversity parameters with respect to the peripheral positions (high, low) was clearly shown (Figure 4i–l).

### 3.3. Dominance of Soil Plant Pathogen Taxa

The proportions of plant pathogenic fungal sequences showed an almost absolute dominance of the Phylum *Ascomycota*, with a very scarce presence (<0.5%) of *Basidiomycota* (Appendix A). Total predominance of the class *Dothideomycota* was observed, as was a presence in almost all sites of the class *Sordariomycota*, and sporadically of *Leotiomycota* (Appendix A). At the order level (Appendix A), a dominance of *Venturiales* was observed at all sites, except in the centers of the gaps, where its dominance was shared with the orders *Pleosporales* and *Hypocreales*. In contrast to the near-total dominance of *Venturiales* in the controls, the application of silvicultural treatments promoted the increase of *Hypocreales* in thinning plots and gap centers, as well as *Pleosporales* in the gaps. The genera (Figure 5) with representation greater than 0.5% were (in order from highest to lowest) as follows: *Venturia* (75%), *Alternaria* (6%), *Gibberella* (5%), *Ilyonectria* (4%), *Mollisia* (3%), *Nectria*, *Microdochium*, *Ophiosphaerella*, and *Drechslera* all with 1%, and finally *Kabatiella*, *Herpotrichia*, *Fusarium*, *Chalastospora*, *Plenodomus*, *Collophora*, *Didymella*, *Phacidium*, *Ceratobasidium* and *Botrytis* with a representation of <1%. Of all these, significant changes (*p* < 0.05) by site effect were only observed in the genera *Venturia*, *Gibberella*, *Ilyonectria*, *Alternaria*, *Kabatiella* and *Fusarium* (Figure 6). Globally (Figure 6a), the application of thinning treatment significantly (*p* < 0.05) promoted the increase of the genus *Ilyonectria* (9%) with respect to the gaps (<2%), although not with respect to the control (2%). The application of gap cuttings caused a significant (*p* < 0.05) increase of the genus *Alternaria* (10%), both with respect to the control and thinning treatments (<2%). The gaps also promoted significantly (*p* < 0.05) the increase of *Gibberella* (9%) with respect to the control (<2%), although not with respect to thinning. In the thinning plots (Figure 6b), the contribution of coarse woody debris significantly (*p* < 0.05) increased the presence of *Fusarium* (0.25%), although its representation was very low with respect to the rest of the main genera. In the gap plots (Figure 6c), the dominance of the genus *Venturia* was significantly reduced (*p* < 0.05) at the central points of the gaps (34%) relative to the extremes (84% and 86%); the representation of *Gibberella* (26%) was significantly increased relative to the high and low edges of the gap (<2%), as was *Kabatiella* (4%) relative to the low edge of the gap (<2%). A heat map shows the relative abundances of the main genera with respect to the total number of plant pathogenic fungi (Figure 7). Within the main genera found, only the following species were identified: *Alternaria_brassicae*, *Alternaria_eichhorniae*, *Alternaria_eureka*, *Alternaria_hungarica*, *Alternaria_kulundii*, *Alternaria_leptinellae*, *Chalastospora_ellipsoidea*, *Fusarium_proliferatum*, *Gibberella_tricincta*, *Ilyonectria_macrodidyma*, *Ilyonectria_radicicola*, *Microdochium_phragmitis*, *Mollisia_cinerea*, *Nectria_ramulariae*, *Ophiosphaerella_herpotricha*, and *Plenodomus_biglobosus*.

### 3.4. Community Structure and Environmental Effect on Soil Plant Pathogens

The structures of plant pathogenic fungal communities were significantly different, both between silvicultural treatments (*p* < 0.001) (Figure 8a; Appendix A) and between thinning intensities (*p* < 0.01) (Figure 8b; Appendix A). However, the block effect could significantly affect these structures (*p* < 0.05; Appendix A). In the gaps (Figure 8c; Appendix A), the internal variability of fungal communities found in the larger diameter gaps (Ø44) made them not significantly different (*p* > 0.05) from the communities found in smaller diameter gaps (Ø26), nor from the controls. The large variability of fungal communities between thinning plots and gaps was primarily caused by the central positions of the gaps (Figure 8a,c). Among thinning plots (Figure 8b; Appendix A) significant differences (*p* < 0.01) were observed between B35 and B20 communities. In any case, the presence of coarse woody debris did not significantly (*p* > 0.05) modify the community structure of plant pathogenic fungi (Appendix A). In the gaps, a significant effect (*p* < 0.05) of central versus peripheral sampling positions was shown, without clearly differentiating the peripheral communities from each other (Figure 8c; Appendix A).

The RDA (Figure 9) showed that much of the fungal communities in gaps were influenced by insolation, altitude, and low ICV, while the thinning communities and controls moved in an average ICV environment.

Negative correlations of ICV with evenness (*p* < 0.01), and of insolation with richness (*p* < 0.05) were observed (Appendix A), consistent with the RDA in Figure 10. On the other hand (Figure 11, Appendix A), positive correlations of ICV with the *Ilyonectria* genus (*p* < 0.01) and of slope with *Mollisia* (*p* < 0.01) were observed; on the other hand, negative correlations of ICV with *Alternaria* (*p* < 0.001), *Gibberella* (*p* < 0.05), and *Kabatiella* (*p* < 0.05) were observed, as were negative correlations of insolation with *Ilyonectria* (*p* < 0.001), *Nectria* (*p* < 0.01), *Drechslera* (*p* < 0.05), and *Ophiosphaerella* (*p* < 0.05), negative correlations of altitude with *Drechslera* (*p* < 0.01), and, finally, negative correlations of slope with *Ilyonectria* (*p* < 0.001) and with *Nectria* (*p* < 0.01).

## 4. Discussion

### 4.1. Abundance of Soil Pathogens

This work showed that some silvicultural treatments significantly modified the communities of plant pathogenic soil fungi in a mature reforestation pine forest.

Soil fungi include pathogens, symbiotic mutualists, and saprotrophic fungi that can promote and maintain tree community diversity through negative or positive plant–soil feedback. Afforestations carried out in the 1970s in Spain, such as the one in the present work, were fundamentally monospecific (Peman-Garcia et al., 2017) [28]; therefore, at least initially, they could generate unstable ecosystems with a higher occurrence of pathogens and pests (Strauss 2001 [29]; Liu & Li, 2010 [30]). Soil-borne pathogens are especially damaging as they infect tree roots, limiting their growth and increasing their susceptibility to other diseases (Yu et al., 2022 [31]). However, the virulence of pathogens depends on many factors, such as environmental conditions, host establishment time at a site, and host and pathogen genotypes (Liu & He, 2019 [32]). In addition, pathogen abundance may decrease with succession (Hannula et al., 2017 [33]). Our work showed that, for this 50-year-old afforestation, the relative abundance of soil pathogenic fungi in 2023 did not exceed 18% and plant pathogens did not exceed 12%, regardless of the silvicultural treatments applied in 2018. Functional balances of soil fungal communities are decisive, not only in the possible harmful effects of pathogenic fungi on plants, but globally in the biogeochemical functioning of the forest soil. In our work, more than 80% of the fungi in the soil corresponded to symbiotic and saprotrophic fungi, so we can consider it to have been balanced fungal edaphic ecosystem at that time.

The application of silvicultural thinning or gap treatments in 2018 generated an environmental disturbance that could have affected soil microbial communities, specifically plant pathogenic fungi, both in structure and species abundance. These disturbances (more or less abrupt or intense for the soil and for fungi), may involve, among other factors, elevation of soil temperature and insolation, modification of soil moisture, or modification of food–host availability. In this work, we observed a significantly higher relative abundance of plant pathogen OTUs in the thinning area than in the gaps (Figure 3a), although no significant differences were observed between thinning plots and controls. We can initially surmise that a wide, abrupt light setting (gaps) could have a restrictive effect on soil plant pathogens versus a more gradual light setting (thinnings); this would be related to the rise in soil temperature and the soil pathogen pool prior to the application of the silvicultural treatments in 2018; on the other hand, and considering the trophic characteristics of soil plant pathogens, a greater contribution of organic leaf litter material in thinning plots—especially needles and pine cones—could promote a greater abundance of plant pathogens with saprotrophic capacity with respect to the gaps. This is consistent with the identification of the main genera of plant pathogenic soil fungi found in our work (Figure 5); most of them were also leaf litter saprotrophs (*Alternaria*, *Fusarium*, *Gibberella*, *Kabatiella*, *Mollisia*, *Nectria*, *Ophiosphaerella*, *Plenodomus*, *Venturia*, *Chalastospora*, etc.). Finally, the presence and diversity of potential plant hosts can define the abundance and diversity of plant pathogens, including those that provide propagules to the soil or are strictly edaphic; in this sense, the application of thinnings or gaps modifies the plant community, and therefore the plant pathogens. In our work, although in year 0 the plant cover in the gaps could be considered null, after 5 years, a plant community composed mainly of regenerated pine and rockrose was established from the center to the gaps’ edges. According to our results, it does not seem that these regenerated plant host communities in the gaps are related to a lower abundance of soil pathogens compared to the thinning plots. In any case, neither thinning intensity, gap diameter, coarse woody debris contribution nor gap sampling position significantly affected the relative abundance of soil plant pathogens (Figure 3b–e), so there might be an opposite relationship between soil pathogen abundance and silvicultural intervention intensity.

### 4.2. Diversity of Soil Plant Pathogens

The opening of the gaps generated an increase in diversity in the centers of the gaps, while strong low thinning (B35) also increased diversity. Only in the centers of the gaps were all the diversity parameters analyzed significantly modified with respect to the edges of the gaps (Figure 4). When comparing treatments between thinning and gaps, the thinnings showed significantly higher pathogen species richness than the gaps and the controls; on the other hand, the gaps showed significantly higher evenness than the thinnings (Figure 4a,d). This shows that, although the thinnings contributed to changes in terms of the number of soil-borne plant pathogen species, the gaps generated the most intense changes in the soil-borne plant pathogen community in the center; this means that the opening of new and large gaps in the vegetation may more effectively facilitate the arrival of new and more distant fungal propagules by dissemination compared to gradual canopy openings, as in thinnings. On the other hand, after 5 years, heliophilous understory vegetation, including new *P. pinaster* seedlings, began to occupy and regenerate the gaps generated by the applied treatments; this regeneration represents new potential hosts for plant pathogens and may also regulate the presence of soil plant pathogens (DeBellis et al., 2007 [34]). In the gaps, regeneration was established with greater intensity/plant diversity in the central zone and more gradually towards the periphery (Figure 9). In addition, abrupt changes in feeding substrates and soil environmental conditions for fungal pathogens occurred in the gaps. Thus, the dominance of the genus *Venturia* on the forest floor was partially broken in the centers of the gaps with the increase of other genera such as *Alternaria*, *Kabatiella*, *Gibberella*, *Nectria*, *Ophiosphaerella*, etc. (Figure 7), all of which are also leaf litter saprotrophs. This may indicate, as we have mentioned, a change in the type of growth substrate or a greater ease of arrival for new fungal propagules to the soil, which are disseminated by wind, rain, insects, or animals. The size of the gap did not significantly affect diversity, although an increasing trend was observed in accordance with the size of the gap, perhaps related to a greater capacity to receive disseminated fungal propagules. On the other hand, in the thinning stands, the strong low thinning (B35) significantly promoted a greater evenness of pathogen species with respect to A35. This seems to indicate that, although thinning intensities or types generated slight differences in soil plant pathogen diversity indices, the stronger and more homogeneous low thinning treatment of B35 allowed the generation of significant changes in the soil plant pathogen community. This is consistent with the NMDS (Figure 8b) and PERMANOVA analysis (Appendix A) performed, which showed significant differences (*p* < 0.01) between the B35 and the B20 communities. In any case, neither the selective thinning of A35 nor the low thinning of B20 significantly modified the soil plant pathogenic fungi community. Similarly, the low impact of selective thinning on the dominant components of soil fungal communities has been observed in other studies (Makipaa et al., 2017 [35]). We can infer that, although A35 may have generated some wider gaps in the canopy after selective removal of dominant trees, B35 generated a more homogeneous openness along the entire sampling transect. Therefore, we can affirm that for this case a 35%-lower thinning is sufficient to generate changes in the soil fungal pathogen community, and that after 5 years the maintenance of coarse woody debris of *P. pinaster* did not affect the diversity or abundance of soil plant pathogens.

### 4.3. Dominance of Soil Plant Pathogen Taxa

In the present study, the dominance of the genus *Venturia* among forest soil plant pathogens is probably the legacy of a situation prior to the application of the 2018 silvicultural treatments. Although *Venturia populina* can cause defoliation of poplar branches and leaves, in our study area the presence of the genus *Populus* is null, so it does not currently pose a threat to woody species in the reforestation area. The presence of root pathogens such as *Gibberella* (*Fusarium* Anam.) and *Ilylonectria* was observed. The high representation of leaf litter saprotrophic plant pathogens means that their survival strategies are effective, even if they do not find plant hosts, provided that adequate levels of leaf litter residues are maintained. The genus *Venturia* is usually more abundant in early stages and has a positive effect in disturbances (Castaño et al., 2018 [36]; Rodríguez-Ramos et al., 2021 [37]; Martin-Pinto et al., 2021 [38]). However, in our work, the application of thinning did not significantly affect the presence of *Venturia*; on the contrary, the gaps significantly decreased its representation in relation to other genera, such as *Alternaria, Gibberella* and *Kabatiella*; *Alternaria* is frequent in Mediterranean oak woodlands and dehesas, and may have an antifungal and biocontrol activity on other plant pathogens (Costa et al., 2020 [39]); *Gibberella circinata* (anamorph *Fusarium circinatum*) can cause pine resin canker, so it would be advisable to monitor it in the future due to its potential threat to *P. pinaster*, the dominant tree species; *Kabatiella prunicola* can cause anthracnose in cherry. On the other hand, the application of thinning allowed a significant increase in the genus *Ilyonectria* (Figure 6a and Figure 7); *Ilyonectria destructans* is a pathogenic fungus causing root rot and other symptoms in trees. Mora-Sala et al. (2018) [40] also reported the occurrence of species belonging to the genera *Ilyonectria* to be associated with seedlings of diverse hosts showing decline symptoms in forest nurseries in Spain: *Nectria cinnabarina* can cause cankers on *Acer*, *Betula*, *Carpinus*, *Fagus*, *Quercus*, *Ulmus*, etc.; *Botrytis cinerea* is a polyphage that can cause blue mold on nursery seedlings; *Botrytis sylvatica* can cause carbonaceous plaque of cork oak; *Herpotrichia juniperi* can attack shoots and needles of conifers. In any case, the representations of most genera were very low.

### 4.4. Community Structure and Environmental Effect on Soil Plant Pathogens

The role of soil plant pathogens may be to regulate plant biodiversity according to the Janzen-Connel principle (Liu et al., 2012 [41]). Although fungal communities, including pathogens, can shape plant community structures and their biodiversity (Winder & Shamoun, 2006 [42]), plant species diversity also, conversely, influences fungal community structures (DeBellis et al., 2007 [34]). In this work, the dominant upper tree cover corresponded to *P. pinaster* established in 1970; a lower understory cover of different woody and herbaceous species may also influence, depending on the site, the structure of soil microbial communities, including of fungi. NMDS showed that the fungal communities of soil plant pathogens in the central areas of the gaps (Figure 8a,c) were different from each other (with a significant block effect) and different from the rest of the sites considered (Appendix A). In any case, no differences were observed between pathogenic fungal communities in the periphery of the gap (Appendix A); this means, contrary to our initial hypothesis, that neither direct insolation nor peripheral position in the gap affected soil pathogen community structure. This gives less weight to the influence of shade on changes in the fungal pathogen community. Gap diameter also did not generate significant changes in soil fungal pathogen community structure (Appendix A). The NMDS (Figure 8b) and PERMANOVA analysis (Appendix A) also showed that communities of thinning B35 were significantly different than those of B20 (*p* < 0.01); they also showed that communities of selective thinning A35 were not significantly different from those of the rest of the thinnings or the control; this shows that the selective thinning A35, despite corresponding to an intensity in volume of wood removed similar to B35, did not generate a significant change in soil pathogens compared to B35, probably because the gaps opened by A35 were not homogeneous along the sampling transect; perhaps other forms of soil sampling could better reflect changes in soil fungal communities for selective thinnings such as A35.

The effect of pathogens on seedling establishment may vary in space, not only in response to the abundance of susceptible hosts, but also in response to heterogeneous abiotic conditions. In our work, despite the lack of soil data, we considered homogeneous macroclimatic values for all sites; we also considered physiography and vegetation cover as two of the main factors that modeled the microclimatic conditions at each site. As observed in the RDA (Figure 9), soil pathogenic fungal communities basically clustered in communities with higher insolation and low aboveground ICV (gaps), or, conversely, in communities with lower insolation and higher aboveground ICV (thinning). In any case, the heterogeneity between blocks affected the differentiation of communities between sites. Fungal species richness (Figure 10; Appendix A) was linked to areas with lower insolation and higher ICV (thinning and control areas), i.e., areas with no or homogeneously small vegetation openings whose pathogen species pool already existed prior to the silvicultural treatments. In contrast, where there was intense opening and sufficiently large gaps, sporal dispersal, sudden change of fungal feeding substrate, and association with new regeneration hosts could increase pathogenic fungal diversity differentially in the centers of the gaps. This is in line with a strong correlation between evenness and absence of ICV cover (Appendix A). Although, as we can see, the level of insolation reaching the soil (defined by a combination of insolation and ICV) was able to modify soil environmental conditions and thus fungal communities (Figure 9), there were some genera of soil plant pathogens that were related to other environmental parameters (Figure 11, Appendix A): *Mollisia,* whose abundance was associated with steeper areas; *Ilonectria* and *Nectria*, associated with flatter areas; *Drechslera*, associated with lower altitude areas; *Alternaria*, *Gibberella*, and *Kabatiella*, associated with low ICV (we already observed that they appeared as soon as the gaps were opened). However, *Ilyonectria*, *Nectria*, *Drechslera*, and *Ophiosphaerella* were associated with areas of low insolation (that which reached the first vegetation layer). Perhaps other environmental parameters, in particular some soil parameters, could explain more completely the distribution of plant pathogenic fungal communities. In any case, and as we have commented above, the representations of most of the genera of soil pathogenic fungi were so low with respect to the rest of the fungal communities that, at present, they do not pose a threat to the vegetation.

## 5. Conclusions

The pathogenic pool of the genus *Venturia* throughout the forest was slightly altered by the silvicultural treatments after 5 years of their application. Only the gap centers showed significant changes in this domain. The opening of large gaps facilitated the entry of new pathogens or increased the abundance of other pre-existing soil pathogens, regardless of the diameter of the gap. After 5 years, these changes were not apparent at the periphery of the gaps. Thinning slightly altered the soil pathogen community as a function of thinning intensity or type; strong low thinning (B35) was required to generate significant changes. Selective thinning (A35) did not reflect a homogeneous change in the soil pathogen community, and maintenance of coarse woody debris also did not generate significant changes in the soil pathogen fungal community. Finally, the large dissimilarity among soil pathogenic fungal communities due to block effects shows the importance of other microenvironmental factors compared to silvicultural treatments, and should be considered in future studies.

## Figures and Tables

**Figure 1 pathogens-13-00637-f001:**
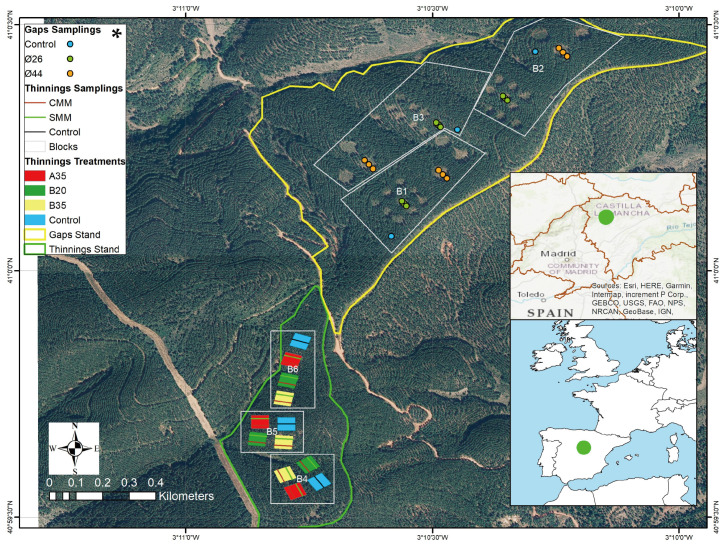
Location and design of experimental area *. Silvicultural treatments (thinning or gap); thinning intensity: strong selective thinning (35% basal area; A35), strong low thinning (35% basal area; B35), and moderate low thinning (20% basal area; B20). Thinning plots with (CMM) or without (SMM) coarse woody debris; gap diameters of 26 m (Ø26) or 44 m (Ø44); gap soil sampling position in High, Center, or Low area. Using ArcMap 10.8.1©, GPSMAP^®^ 66i, and digital cartography by the National Geographic Institute of Spain. https://www.scne.es/ (accessed on 20 April 2024).

**Figure 2 pathogens-13-00637-f002:**
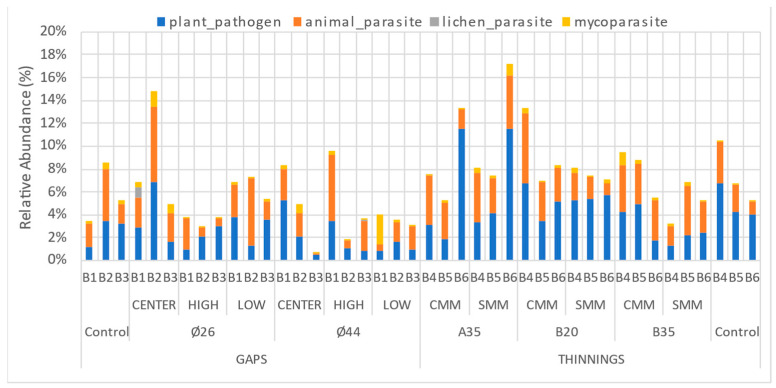
Relative abundance (OTUs) of soil plant pathogen fungi according to silvicultural treatments (thinning or gap), thinning intensity (A35, B35, B20), status of with (CMM) or without (SMM) coarse woody debris in thinning plots, gap diameters (Ø26 or Ø44), and gap sampling position (High, Center, Low).

**Figure 3 pathogens-13-00637-f003:**
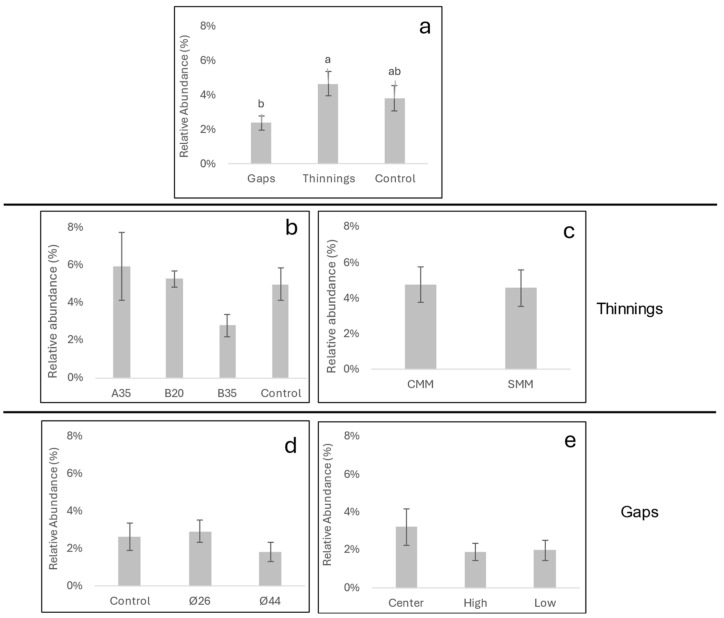
Comparison of relative abundance (OTUs) of soil plant pathogen fungi according to the following: (**a**) silvicultural treatments; (**b**) thinning intensity; (**c**) status of with (CMM) or without (SMM) coarse woody debris in thinning plots; (**d**) gap diameters; (**e**) gap sampling position. Bars indicate individual standard error. Different letters indicate significant differences at *p* < 0.05 according to Tukey’s, Duncan, and Kruskal-Wallis Tests, or n.s. for not significant. N = 6 and 18 (**a**), 3 and 6 (**b**), 9 (**c**), 3 and 9 (**d**), and 6 (**e**).

**Figure 4 pathogens-13-00637-f004:**
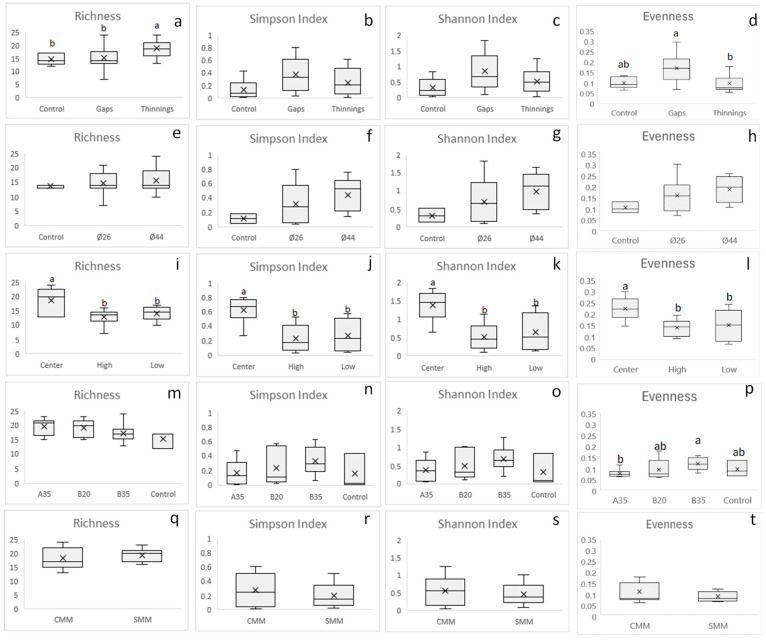
Box-plots of the richness, diversity index and evenness of total plant pathogen fungi sequences (OTUs) according to silvicultural treatments (thinning or gap; (**a**–**d**)), thinning intensity (A35, B35, B20; (**m**–**p**)), status of with (CMM) or without (SMM) coarse woody debris in thinning plots (**q**–**t**), gap diameters (Ø26 or Ø44; (**e**–**h**), and gap sampling position (high, center, low; (**i**–**l**)). Bars indicate individual standard error. Different letters indicate significant differences at *p* < 0.05 according to Tukey’s, Duncan, and Kruskal-Wallis tests. N = 6 and 18 (**a**–**d**), 3 and 9 (**e**–**h**), 6 (**i**–**l**), 3 and 6 (**m**–**p**), and 9 (**q**–**t**).

**Figure 5 pathogens-13-00637-f005:**
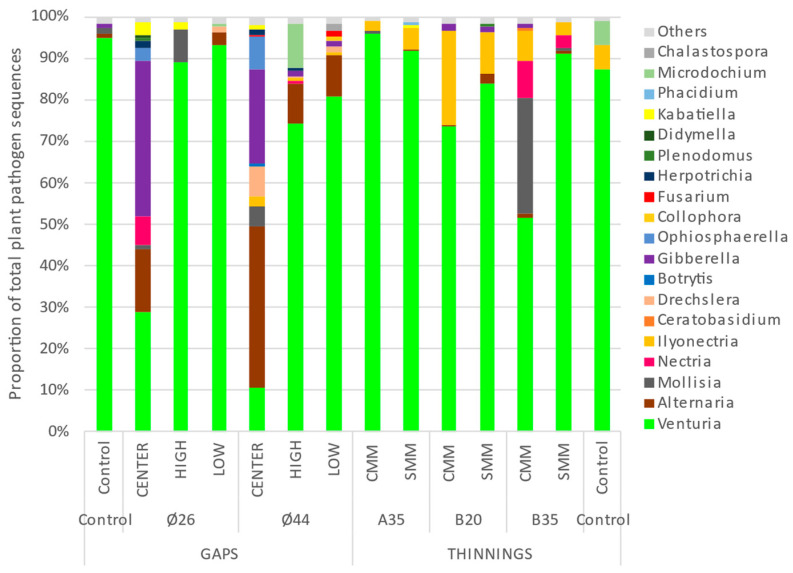
Genus proportion average of total plant pathogen fungi sequences.

**Figure 6 pathogens-13-00637-f006:**
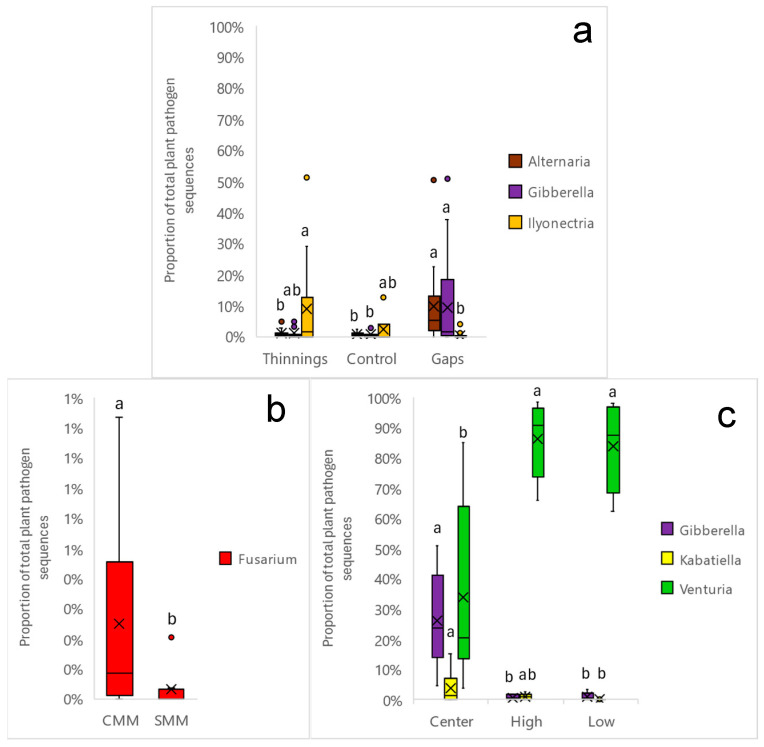
Box-plots of main Genus proportion of plant pathogenic fungi that have shown significant changes due to different silvicultural treatments (thinning or gap), status of with (CMM) or without (SMM) coarse woody debris in thinning plots, and gap sampling position (center, high, low). For each color, bars indicate individual standard error. Different letters indicate significant differences at *p* < 0.05 according to Tukey’s, Duncan, and Kruskal-Wallis tests. N = 6 and 18 (**a**), 9 (**b**), and 6 (**c**).

**Figure 7 pathogens-13-00637-f007:**
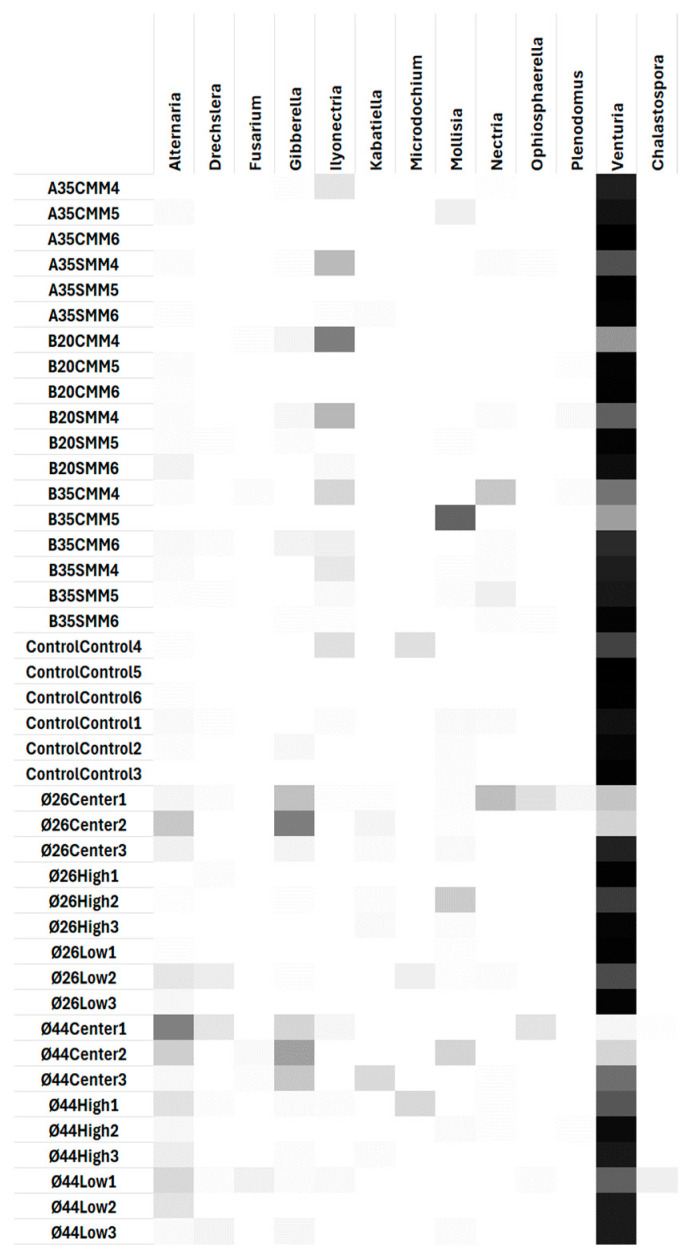
Heat map of main genus rate of soil plant pathogenic fungi according to silvicultural treatments (thinning or gap), thinning intensity (A35, B35, B20), status of with (CMM) or without (SMM) coarse woody debris in thinning plots, gap diameters (Ø26 or Ø44), and gap sampling position (center, high, low).

**Figure 8 pathogens-13-00637-f008:**
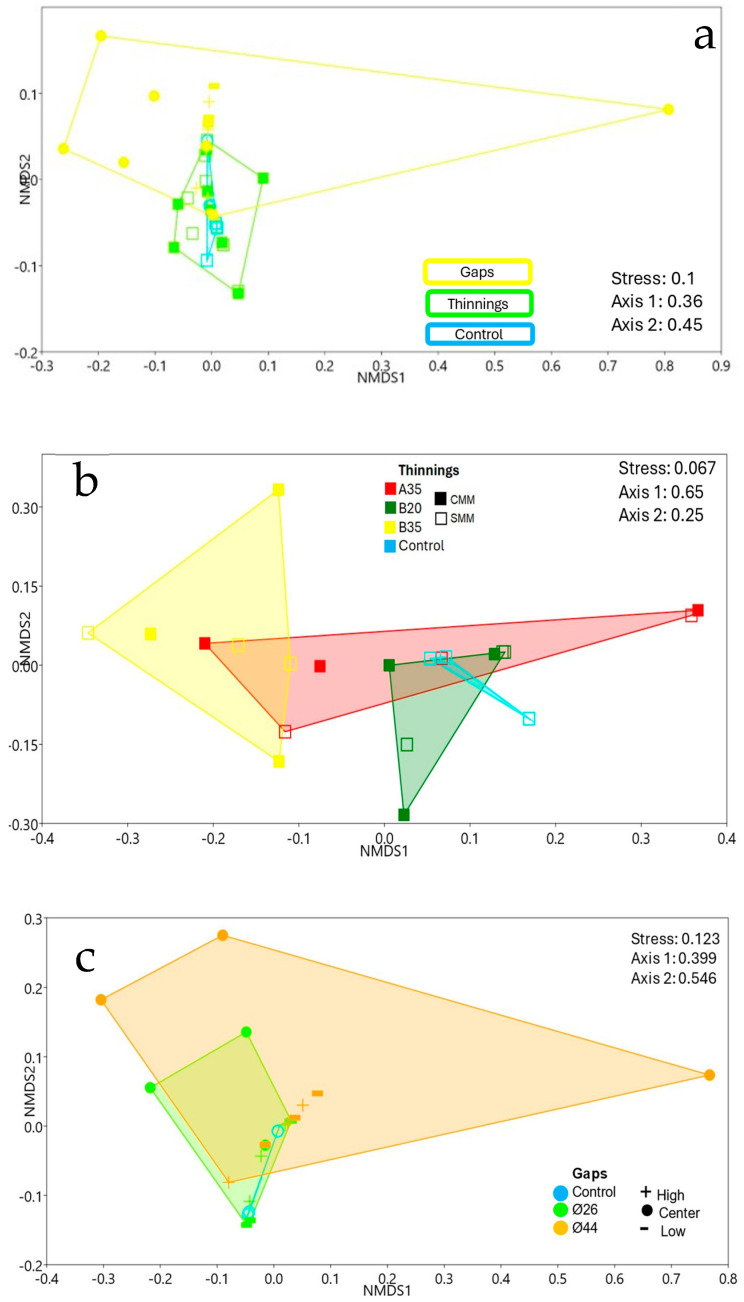
NMDS of fungal communities according to (**a**) treatments, (**b**) thinning intensity, and (**c**) gap diameters.

**Figure 9 pathogens-13-00637-f009:**
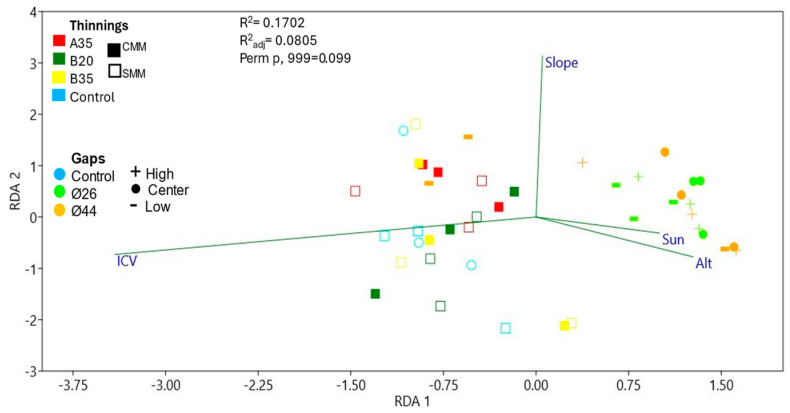
Redundance Analysis of fungal communities and environmental variables.

**Figure 10 pathogens-13-00637-f010:**
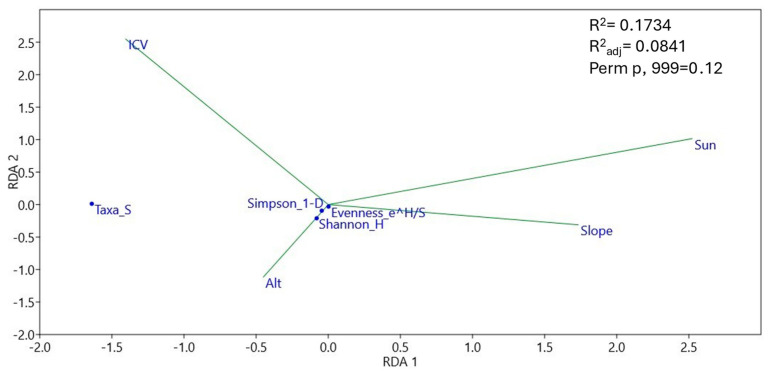
Redundance analysis of diversity and environmental variables.

**Figure 11 pathogens-13-00637-f011:**
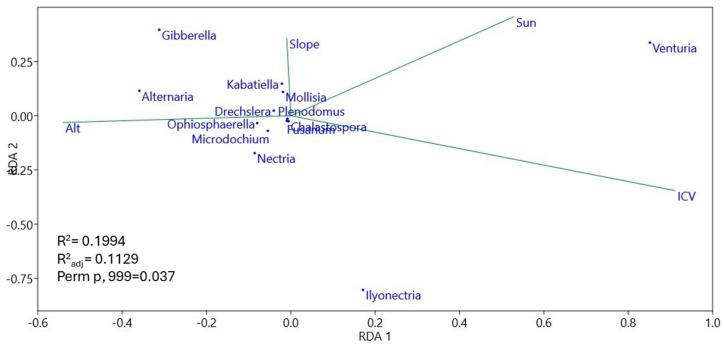
Redundance analysis of main fungal genus and environmental variables.

## Data Availability

The raw data supporting the conclusions of this article will be made available by the authors on request.

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
