# Peer review of "Soil Fungal Pathogens in Pinus pinaster Mature Reforestation: Silvicultural Treatments Effects"

_pathogens, 2024, doi:10.3390/pathogens13080637_

Round 1

Reviewer 1 Report

Comments and Suggestions for Authors

The manuscript of this paper has shown scientific and readable performance, with clear research objectives, reasonable design, accurate methods, and potential for publication in journals. However, there are some deficiencies in the logical coherence of the results presentation and discussion sections, and the clarity of some images needs to be improved.

1. Regarding the presentation of the results section, it is recommended that the author subdivide the content of the results into 2 to 3 core themes and add clear subheadings for each theme. This structure helps readers understand the research results more clearly and also provides a more specific framework for the discussion section.

2. The discussion section should closely revolve around the content of the results section. Authors need to ensure that the discussion corresponds to the main results, providing in-depth analysis and explanations for each core result. This not only helps to improve the logic and coherence of the paper, but also enables readers to more fully understand the value and significance of the research.

3. Authors should carefully check the resolution and clarity of the images to ensure that they accurately reflect the research results. If necessary, appropriate processing or replacement can be made to improve their readability and comprehensibility.

In summary, the manuscript of this paper exhibits excellent overall quality, but there is still room for improvement in terms of the presentation of results, the logical coherence of the discussion section, and the clarity of the images. By making improvements and refinements on the concerns mentioned above, the manuscript will become more competitive and more likely to be accepted.

Reviewer 2 Report

Comments and Suggestions for Authors

The study is interesting but it is only descriptive. I think that the richness of the work is great for how little the data is being exploited.

 The discussion of results is not specific to the study but rather they say that perhaps other authors mention that I think they should be focused on discussing what was observed in the study.

The number of samples, their degree of representativeness, and the sampling density per block must be stated. Line 164, what size is the sample plot?

Improve resolution of graphics and figures

Improve writing and focus on the results of the experiment

Write a conclusion so that it is a conclusion and not just a description of what was observed and correlate it to the different silvicultural treatments or say specifically that it does not represent any correlation.

It is noted that the person who wrote the article has a specialization in microbiology. A general suggestion is that he relies on his other collaborators and they help him with the statistical part and geographic information systems, to describe the stratification of the environment and the representativeness of the data

Round 2

Reviewer 2 Report

Comments and Suggestions for Authors

I consider the changes to be sufficient to have the minimum to be published